# The Protective Effects of Iron Free Lactoferrin on Lipopolysaccharide-Induced Intestinal Inflammatory Injury via Modulating the NF-κB/PPAR Signaling Pathway

**DOI:** 10.3390/foods11213378

**Published:** 2022-10-26

**Authors:** Hongya Wu, Linlin Fan, Yanan Gao, Jiaqi Wang, Nan Zheng

**Affiliations:** 1Key Laboratory of Quality & Safety Control for Milk and Dairy Products of Ministry of Agriculture and Rural Affairs, Institute of Animal Sciences, Chinese Academy of Agricultural Sciences, Beijing 100193, China; 2Laboratory of Quality and Safety Risk Assessment for Dairy Products of Ministry of Agriculture and Rural Affairs, Institute of Animal Sciences, Chinese Academy of Agricultural Sciences, Beijing 100193, China; 3Milk and Dairy Product Inspection Center of Ministry of Agriculture and Rural Affairs, Institute of Animal Sciences, Chinese Academy of Agricultural Sciences, Beijing 100193, China; 4State Key Laboratory of Animal Nutrition, Institute of Animal Sciences, Chinese Academy of Agricultural Sciences, Beijing 100193, China

**Keywords:** iron free lactoferrin, protective effects, intestinal inflammatory injury, NF-κB/PPAR signaling pathway

## Abstract

Research evidence shows that effective nutritional intervention could prevent or reduce intestinal inflammatory injury in newborn infants. Iron free lactoferrin (apo-LF), one of the main types of lactoferrin (LF), is a bioactive protein in milk that plays a vital role in maintaining intestinal health. The potential mechanism by which apo-LF modulates intestinal inflammation is, however, still unclear. In the study we first explored key genes and pathways in vitro by transcriptome date analysis and then validated them in vivo to reveal the underlying molecular mechanism. The results showed that apo-LF pretreatment effectively inhibited lipopolysaccharide (LPS)-induced primary intestinal epithelial cells (IECs) inflammation in the co-culture system (primary IECs and immune cells), which was specifically manifested as the reduction of the concentration of TNF-α, IL-6 and IL-1β and increased the concentration of IFN-γ. In addition, transcriptome data analysis revealed that the key pathway for apo-LF to exert anti-inflammatory effects was the NF-κB/PPAR signaling pathway. Further validation was performed using western blotting in colonic tissues of young mice and it was found that the major proteins of NF-κB signaling pathway (NF-κB, TNF-α and IL-1β) were inhibited by apo-LF and the target proteins of PPAR signaling pathway (PPAR-γ and PFKFB3) were activated by apo-LF. Taken together, this suggests that apo-LF has a protective effect against LPS-induced intestinal inflammatory injury via modulating the NF-κB/PPAR signaling pathway, which provides new insights for further anti-inflammatory study of apo-LF.

## 1. Introduction

Against intestinal pathogens, the mucosal immune system acts as a first line of defense for the body, consisting of closely arranged epithelial cells, immune cells and their secretions and providing a selective permeable barrier [1]. Studies have shown that intestinal mucosa is not only an important place for nutrient absorption, but also the main site of infection by many pathogens. Once damaged, it will lead to tissue damage and systemic inflammation [2]. Different from adults, the transition from the warm environment of a mother’s womb to the dangerous outside world is a huge challenge for newborn infants. In particular, the intestinal tract in the immature stage is vulnerable to damage under external environmental stimuli (such as stress, infection and pathogen invasion), which can induce intestinal diseases including necrotizing enterocolitis and inflammatory bowel disease [3,4,5]. This suggests that, for newborns, it is urgent to establish a strong intestinal immune system to resist external hazards. 

Milk provides a variety of active proteins with anti-inflammatory properties, among which lactoferrin (LF) is the most widely studied. Research evidence makes clear that LF could promote the establishment and rapid development of the intestinal immune barrier in newborn individuals and help newborn individuals defend against inflammatory diseases [6,7]. According to the different amounts of iron binding, LF is divided into three main forms: apolactoferrin (apo-LF, iron-free), a mono-ferric form (containing one iron ion) and hololactoferrin (holo-LF, containing two iron ions). However, due to the limited technology of its separation and extraction, only apo-LF and holo-LF are commercially sold at present. Study has demonstrated that LF with different iron saturation has significant differences in inhibiting macrophage inflammation, among which apo-LF has the strongest inhibitory effect [8]. In addition, the dietary supplementation of different iron saturation LF in Kenyan infants found that apo-LF binds iron with high affinity and promotes iron absorption in infants [9]. Acting on the intestine, apo-LF could stimulate human colon cancer cell Caco-2 proliferation [10] and inhibit rat ileal cell IEC-18 migration [11], effectively alleviating ileal tissue damage [12]. Taken together, the study demonstrated that apo-LF possessed more powerful biological functions than the other types of LF.

Utilizing transcriptomics techniques can achieve an unprecedented level of detail in understanding cellular phenotypes by examining genes expressed in specific physiological and pathological states [13]. Currently, the application of transcriptomic technology to explore the effect of functional food supplementation on intestinal inflammation is very mature. Shin et al. used ileal transcriptome analysis and found that *Lactobacillus plantarum* strain JDFM LP11 reduced the expression of ileal immune genes on intestinal inflammation and promoted the intestinal development of weaned piglets [14]. Furthermore, colonic transcriptomic analysis of mice revealed that dietary galacto-oligosaccharides could reduce intestinal permeability and increase mucus production by modulating changes in host gene expression [15]. In addition, owing to the fact that the intestinal microenvironment is frequently altered, primary cell culture becomes a promising and effective in vitro model for understanding intestinal epithelial function [16]. Immune cells interact with epithelial cells in the intestine to jointly maintain the function of the intestinal immune barrier. The traditional in vitro model of intestinal epithelial cells (IECs) does not consider the existence of immune cells, which also causes some limitations in understanding the function of the intestinal immune system [17]. However, there are few reports on the effect of apo-LF on neonatal intestinal inflammation by using transcriptomics technology and there are relatively few studies on the co-culture system (primary IECs and immune cells) of apo-LF. 

Based on previous findings, this study used transcriptomics technology to reveal the protective mechanism of apo-LF on lipopolysaccharide (LPS)-induced primary IECs inflammation of young mice in the co-culture system and verified this in colonic tissue of young mice, in order to further expand the related function of apo-LF on neonatal intestinal inflammation injury.

## 2. Materials and Methods

### 2.1. Reagents

The specific preparation method of apo-LF is as described in the previous patent *A method for preparing lactoferrin with required iron saturation* [18]. Gibco (Carlsbad, CA, USA) provided Dulbecco’s modified Eagle’s medium (DMEM)/F-12 and fetal bovine serum (FBS). Beijing Solarbio Technology Co. Ltd. (Beijing, China) provided LPS, ELISA kit for mouse TNF-α, mouse IL-6, mouse IL-1β and mouse IFN-γ, primary antibody (PFKFB3, NF-κB, TNF-α and β-actin) and Goat Anti-Rabbit IgG/HRP. Primary antibody (PPAR-γ and IL-1β) was obtained from Bioss (Beijing, China). Protease inhibitor cocktail, penicillin-streptomycin solution, phosphate-buffered saline (PBS), BCA protein assay kit, trypsin-EDTA solution with phenol red and RIPA lysis buffer were provided by Beyotime Biotechnology (Shanghai, China). Difco^TM^ skim milk was obtained from BD-Pharmingen (San Diego, CA, USA).

### 2.2. Culture of Primary IECs

The isolation and culture methods of primary IECs from the colonic tissue of young mice were referred to in previous studies [19]. Intestine tissue was washed in pre-cooled PBS. Next, intestine tissue was cut longitudinally and the colonic epithelial mucosa was scraped off and transferred to preheated digestive solution, which was digested in a water bath for 20 min at 37 °C before being terminated with FBS. The mixture was collected after filtration using cell filters, centrifuged for 15 min at 1200× *g*, then removed the supernatant, repeated twice. Then the cells were inoculated into cell dishes and grown at 37 °C, 95% humidified atmosphere with 5% CO_2_. After the cells adhered, the fibroblast area was scraped off and the IECs were retained for collection. After resuspension in DMEM/F12 medium, the cell precipitation was transferred to a new culture dish for further culture. The above steps were repeated until almost all epithelial cells were observed. 

After 48 h of culture, the isolated primary IECs of young mice showed vigorous growth, with a density of more than 90% and a typical cobblestone arrangement (Appendix A). Subsequently, hematoxylin and eosin (HE) staining were performed and the cell tissue was found to be clearly visible (Appendix A). The preparation of the cell glass slide showed positive results after immunohistochemical identification of mouse cytokeratin18 (Abcam, Cambridge, UK), indicating that primary IECs of young mice were successfully isolated (Appendix A).

### 2.3. Co-Culture and Treatment of Primary IECs and Immune Cells

In order to better simulate the intestinal environment, we co-cultured primary IECs and immune cells in a trans-well chamber at a ratio of 1:1. Procell Life Science and Technology Co., Ltd. (Wuhan, China) provided mouse B cells as immune cells. Mouse B cells were suspended in the lower solution and primary IECs were placed in the upper solution. Four treatment groups were set up, control group, apo-LF alone treatment group, LPS alone treatment group, and apo-LF and LPS combined treatment group. According to a previous study, we chose the concentrations of apo-LF and LPS [20]. The upper chamber was supplemented with FBS-free DMEM/F12 medium containing 10 g/L apo-LF, 1 mg/L LPS and 10 g/L apo-LF + 1 mg/L LPS, respectively. The control group was FBS-free DMEM/F12 medium without any treatment. 48 h after treatment, supernatant from the upper chamber was fractionated and used for subsequent ELISA experiment and cells of each treatment group were collected by the Trizol method for transcriptome sequencing and RT-qPCR verification.

### 2.4. Colon Collection of Young Mouse Models

Beijing Vital River Laboratory Animal Technology Co., Ltd. (Beijing, China) supplied 32 2-week-old healthy young male C57BL/6J mice (10 ± 3 g). Mice were fed at room temperature of 23 ± 1°C and relative humidity of 50 ± 5% for 12 h day-night cycle. The young mice were divided into 4 groups according to the principle of randomization, each containing 8 mice, including control group (gavage of sterile water), apo-LF alone treatment group (gavage of 100 mg/kg b.w. apo-LF), LPS alone treatment group (gavage of 10 mg/kg b.w. LPS) and apo-LF + LPS combined treatment group (gavage of 100 mg/kg b.w. apo-LF + 10 mg/kg b.w. LPS). The whole experiment lasted for 14 days. The mice were killed by cervical dislocation and their colon tissue was collected for further study on the 15th day. The Chinese Academy of Agricultural Sciences (Beijing, China; permission code: IAS202104) approved the entire animal experiment, which is in line with internationally recognized principles for the care and use of experimental animals (NRC, 2001).

### 2.5. Inflammatory Cytokine Determination

48 h before cell collection, cells were grown in DMEM/F12 medium without FBS. After collecting the IECs supernatant in the trans-well upper chamber, the concentrations of factors (IL-1β, IL-6, TNF-α and IFN-γ) were determined by mouse ELISA kit. The solution’s absorbance was measured at 450 nm.

### 2.6. Total RNA Extraction, Transcriptome Sequencing 

As described in previous studies, transcriptome analysis was conducted [21]. Briefly, total RNA was extracted from primary IECs of control treatment group, apo-LF alone treatment group, LPS alone treatment group and apo-LF and LPS combined group by Trizol method. After quality testing of the extracted RNA, eukaryotic mRNA was enriched and subsequently reverse transcribed into cDNA. Transcriptome sequencing was performed using Illumina HiSeq™ 2500 (Illumina, San Diego, CA, USA) after DNA library construction. The identification of differentially expressed genes (DEGs) between different treatment samples was achieved using a fold change > 2 and false discovery rate (FDR) < 0.05. By comparing with human genome, Gene Ontology (GO) enrichment analysis was carried out on DEGs. The biological pathway of DEGs were enriched in the Kyoto Encyclopedia of Genes and Genomes (KEGG). To reduce the differences between groups, three samples were prepared for each cell treatment group and each sample consisted of three replicates.

### 2.7. RT-qPCR Validation

The selected genes were subjected to RT-qPCR to validate the transcriptome data. First, RNA from different treatment groups of primary IECs was extracted by Trizol method and then extracted RNA was reverse transcribed into cDNA with the help of the Prime Script RT Reagent Kit (TaKaRa, Shiga Prefecture, Japan). Primers used for gene quantification were from Shanghai Sanggen Biotech Co., Ltd. (Shanghai, China) and the sequences are shown in Appendix A. Following that, cDNA amplification was completed with the help of a PCR reaction kit (TaKaRa, Shiga Prefecture, Japan). The reaction conditions were denaturation at 95 °C for 1 min, 39 cycles of denaturation at 95 °C for 10 s and annealing at 60 °C for 30 s. The reference gene selected was mouse GAPDH and the results were normalized and calculated by 2^−∆∆Ct^ method.

### 2.8. Western Blotting (WB) 

Colonic tissue (about 2 cm) of young mice was cooled in liquid nitrogen in advance and ground into powder. 150 µL RIPA Lysis Buffer and 2 µL Protease inhibitor cocktails were added to every 10 mg of tissue, and homogenized for 5 min, followed by 15 min on ice. After repeating 3 times, the supernatant was centrifuged. After detecting and quantifying the concentration of protein samples using the BCA assay kit, they were added to 12% SDS-polyacrylamide gels for electrophoresis, followed by electroblotting on polyvinylidene fluoride membranes. Incubation of the membranes in 5% blocking buffer (skimmed milk) followed at room temperature for 1.5 h, followed by incubation with primary antibody (PFKFB3, PPAR-γ, NF-κB, TNF-α, IL-1β and β-actin) at room temperature for 2 h and transfer to 4 °C overnight. After washing the membranes 3 times, the required secondary antibody was added and incubated for 1 h and completed the next day. Band density analysis of the imaged membranes was performed using ImageJ 2 × software (version 2.1.0, National Institutes of Health, Bethesda, MD, USA).

### 2.9. Statistical Analyses

Data analysis was performed using GraphPad Prism 8.0 (GraphPad Software, San Diego, CA, USA). Differences were statistically examined by one-way ANOVA tests followed by Tukey’s multiple comparison test. *p* < 0.05 was considered as statistically significant in this study. In addition, each experiment was independently repeated at least three times. 

## 3. Results

### 3.1. Protective Effect of Apo-LF on LPS-Induced Cellular Inflammation

First, the concentration of inflammatory cytokines in the supernatant of primary IECs of young mice was detected by ELISA kit. As shown in Figure 1A–C, compared with the control group, the concentrations of pro-inflammatory cytokines, specifically TNF-α, IL-6 and IL-1β in the LPS alone treatment group, were significantly increased (*p <* 0.05); after apo-LF intervention, the concentration of the above pro-inflammatory cytokines was still higher than that of the control group, but significantly decreased compared to that of the LPS group (*p <* 0.05). IFN-γ, an anti-inflammatory cytokine, showed the opposite trend (Figure 1D). Compared with LPS alone group, the concentration of IFN-γ increased significantly after apo-LF and LPS combined treatment (*p <* 0.05) and there was no significant difference compared with the control group. LPS induced cellular inflammation and apo-LF supplementation in advance could effectively inhibit the inflammatory response induced by LPS, although they would not completely reverse it, which was proved in the study results above.

### 3.2. Apo-LF Effects on the Intestinal Gene Expression Induced by LPS

In order to explore the mechanisms related to the above phenotypic changes, we used DESeq2 to perform differential analysis of transcriptome data and screen out 3861 DEGs between groups. Among them, there were 1195 DEGs (529 up-regulated and 666 down-regulated) between the apo-LF alone treatment group and the control group (Figure 2A). 1310 DEGs (634 up-regulated and 676 down-regulated) were included in the LPS alone treatment group (Figure 2B). 1170 DEGs (405 up-regulated and 765 down-regulated) were found in the apo-LF + LPS group (Figure 2C). Furthermore, compared with LPS group, there were 186 DEGs in apo-LF + LPS combined treatment group, including 178 up-regulated DEGs and eight down-regulated DEGs (Figure 2D). Cluster analysis of all DEGs showed that three samples in the same group formed independent gene clusters, showing good repeatability and there were obvious differences between the three treatment groups (apo-LF, LPS, apo-LF + LPS) and the control group (Figure 2E). Further analysis of DEGs among the above four comparison groups shows that 156 DEGs are unique to the apo-LF group, 237 DEGs are unique to the LPS group, 178 DEGs are unique to the apo-LF + LPS group and 103 DEGs are unique to the LPS group compared with the apo-LF + LPS group (Figure 2F).

### 3.3. Functional Enrichment Analysis of GO and KEGG Pathway

Subsequently, to understand the detailed protective mechanism of apo-LF against LPS-induced inflammation injury, a total of 52 DEGs related to inflammation were screened from control treatment group, apo-LF alone treatment group, LPS alone treatment group and apo-LF + LPS combined group (Appendix A) and performed GO terms and KEGG enrichment analysis on these 52 DEGs. As seen in Figure 3A, the top five GO terms were inflammatory response, defense response, response to cytokine, response to lipid and response to lipopolysaccharide. The KEGG enrichment analysis revealed that multiple pathways related to inflammation were significantly enriched, including TNF signaling pathway, NF-kappa B (NF-κB) signaling pathway, IL-17 signaling pathway, Cytokine-cytokine receptor interaction, PPAR signaling pathway and NOD-like receptor signaling pathway (Figure 3B). In addition, we also performed KEGG enrichment analysis on 103 unique DEGs screened from the above LPS group compared with the apo-LF + LPS group (Appendix A) and found that they could also enrich the pathways related to inflammation, including the well-known NOD-like receptor signaling pathway, ErbB signaling pathway and Inflammatory bowel disease. The above studies fully demonstrated that apo-LF could ameliorate LPS-induced intestinal inflammatory injury through a variety of inflammatory pathways.

### 3.4. Effect of Apo-LF on LPS-Induced Cellular Inflammatory Gene Expression

According to the functional prediction and targeting of key genes enriched in KEGG, we speculated that the inflammatory pathways playing a key role may be NF-κB and PPAR signaling pathway. Based on this, we randomly selected six DEGs that played roles in NF-κB and PPAR signaling pathways from the above 52 inflammatory related DEGs to verify the reliability of transcriptome data. RT-qPCR results of the 6 DEGs were the same as RNA-seq, as seen in Figure 4. The mRNA expression levels of *Tnfaip3*, *Nfkbia*, *Relb*, *Scd2*, *Bcl3* and *Ccl2* were significantly different (*p* < 0.05) between LPS alone treatment group and control group. After the intervention of apo-LF, the mRNA expression levels of six DEGs showed a downward trend, among which the mRNA levels of *Relb*, *Bcl3* and *Ccl2* decreased significantly (*p <* 0.05). At the same time, we noted that the mRNA levels of *Nfkbia, Relb*, *Scd2* and *Bcl3* were still significantly different between apo-LF + LPS group and control group (*p <* 0.05), indicating that, although apo-LF could inhibit LPS-induced inflammatory response, this did not completely reverse.

### 3.5. Apo-LF Inhibited LPS-Induced Colonic Tissue Inflammation in Young Mice

Finally, to further confirm whether apo-LF plays a key protective role in intestinal inflammatory injury through NF-κB/PPAR signaling pathway, the protein expression levels of PPAR-γ, PFKFB3, NF-κB, TNF-α and IL-1β in colonic tissue of young mice were detected. The relevant results were shown in Figure 5; the protein expression levels of PPAR-γ and PFKFB3 decreased significantly (*p <* 0.05), while the protein expression levels of NF-κB, TNF-α and IL-1β increased significantly (*p <* 0.05) in LPS treatment group compared with the control group. After apo-LF intervention, the above trend could be significantly reversed. In LPS + apo-LF combined group, the protein expression levels of PPAR-γ, PFKFB3 were significantly increased (*p <* 0.05) and the protein expression levels of NF-κB, TNF-α and IL-1β decreased significantly (*p <* 0.05). The western blotting results further verify that apo-LF protected LPS-induced intestinal inflammatory injury through NF-κB/PPAR signaling pathway.

## 4. Discussion

Many infant diseases are intrinsically caused by inflammation and dysregulation of bacterial colonization and the uncontrolled immune response of IECs may aggravate the inflammatory response and the degree of intestinal mucosal destruction [22,23]. An integrated intestinal barrier is an important feature of neonatal health. LF is the most characteristic dietary bioactive protein with multifunctional bioactivity. Several researchers have indicated that it has a positive impact on the intestinal health and overall well-being of infants [24,25,26]. However, few studies specifically involve the impact of apo-LF on the intestinal health of newborn infants. Based on this, we focused on the effects of apo-LF on LPS-induced intestinal inflammation injury of young mice and explored the underlying key molecular mechanisms.

In the present study, primary IECs was successfully isolated from the colonic tissue of young mice and co-cultured with immune cells (Appendix A). Studies have found that the lack of primary cell culture models has become a limiting factor in the study of intestinal epithelial function [27], which has led many researchers to use cancer-derived cell lines to study the physiological function of intestinal epithelium [28,29]. When primary cells leave the body, their biological characteristics are close to those of the original cells and many of the main features and functions of the original tissue are retained. In contrast, the continuous passage of cancer cells is prone to genetic and phenotypic changes, which are inadequate in accurately reflecting in vivo physiology [30]. Previous studies have shown that primary colonic IECs have a greatly different response to inflammation compared with other cancer cell lines when studying colitis, as shown in the fact that IFN-γ can induce the secretion of IL-8 in Caco-2 cells to remain unchanged, while it increases in HT-29 cells and decreases in primary IECs [31]. This further reminds us of the necessity of using the primary IECs model. However, this study more realistically simulated the intestinal environment, different from previous studies.

Cytokines are widely known to have an irreplaceable effect in the intestinal immune barrier [32]. Some of these cytokines have anti-inflammatory properties (IFN-γ, IL-4 and IL-10), while others have pro-inflammatory properties (TNF-α, IL-6, IL-8 and IL-1β). Regulation of LF on its expression has been supported in a number of in vivo and in vitro studies [33,34]. As is known to all, LPS is an inducer of inflammatory bowel disease and intestinal barrier dysfunction, which can induce morphological, metabolic and gene expression changes of IECs [35,36], leading to uncontrolled expression of host cytokines [37,38]. Subsequently, the ELISA method was used to detect cytokines in primary IECs of young mice. It was found that LPS treatment not only significantly induced the secretion of TNF-α, IL-6 and IL-1β, but also led to a significant decrease in the concentration of IFN-γ (*p <* 0.05). Apo-LF treatment significantly reversed these trends and alleviated LPS-induced inflammation (Figure 1). These results were consistent with previous findings in porcine small IECs IPEC-J2 [39] and rat IECs [7]. 

To fully understand the complex development process of some diseases, researchers tend to elaborate the pathological mechanism by comparing the differences in gene expression between different samples to provide evidence for early prevention [40]. We further analyzed the protective mechanism of apo-LF against LPS-induced primary IECs inflammation using transcriptomics and found that DEGs were enriched in the TNF signaling pathway, NF-κB signaling pathway and PPAR signaling pathway, closely related to cellular inflammation (Figure 3B). It was suggested that the specific mechanism of intestinal protective effect of apo-LF may be realized through the NF-κB/PPAR signaling pathway. Relb is one of the more unusual members of the NF-κB family and is known for its role in lymphatic development, dendritic cells biology and noncanonical signaling [41]. Previous studies have confirmed that Relb overexpression in immune cell lines leads to stronger activation of NF-κB and enhanced LPS-induced production of TNF-α and IL-6 [42]. Bcl3 plays a major role in the regulation of immunity and inflammation, promoting or inhibiting the transcription of NF-κB dependent genes, combining with different subunits to exert pro-inflammatory and anti-inflammatory effects [43]. LPS stimulation increased Bcl3 expression and accumulation in the nucleus [44]. Ccl2 is a chemokine that responds to different inflammatory stimuli, such as LPS and TNF-α and can prevent a variety of inflammatory diseases by gene deletion and antibody elimination [45]. In addition to confirming the reliability of transcriptome data, we found that the mRNA expression levels of the above three inflammation related genes (*Relb*, *Bcl3* and *Ccl2*) were significantly increased after LPS stimulation (*p <* 0.05) and decreased significantly after apo-LF intervention (*p <* 0.05), which further confirmed the anti-inflammatory effect of apo-LF. Interestingly, we also noticed that, compared with the control group, *Relb*, *Bcl3* and *Ccl2* were significantly increased to different degrees after apo-LF treatment alone (*p <* 0.05). Previous studies have revealed that cells treated with bovine LF have a strong innate immune response and produce an inflammatory reflect compared with the control group [46]. It is speculated that the increased expression of inflammatory genes may be due to the difference of immune regulation of apo-LF, which is affected by various factors such as treatment time and concentration and mainly plays a related protective role in the stimulation characterized by excessive inflammatory response [47]. Apo-LF inhibited the NF-κB signaling pathway, specifically manifested by reducing the expression level of key proteins (NF-κB, TNF-α and IL-1β) and activated the PPAR signaling pathway, specifically manifested by the increase in the expression of target proteins (PPAR-γ and PFKFB3), as shown in Figure 6. 

As is well known, NF-κB is one of the typical pro-inflammatory signaling pathways, mainly due to its role in the expression of cytokines, chemokines and adhesion factors [48]. After being stimulated by cytokines, oxidative stress and viruses, activated NF-κB may lead to uncontrolled expression of multiple genes, which are involved in innate and adaptive immune regulation and inflammatory response [49,50]. Therefore, inhibiting the activation of the NF-κB signaling pathway is an effective means to control the inflammatory response in the body. Previous studies have shown that dietary bovine LF provides limited protection in the intestine by reducing the protein level of pro-inflammatory transcription factor NF-κB and regulating the expression of cytokines TNF-α, IL-1β and IL-10 [51,52]. Consistent with previous findings, we found that apo-LF significantly inhibited the protein expressions of NF-κB, TNF-α and IL-1β in the colonic tissue of LPS-induced young mice (*p <* 0.05) (Figure 5). PPAR-γ is the most common target among PPAR family members to control immune homeostasis, which can regulate macrophage polarization as well as adaptive immune cells [53], showing a general down-regulation trend in inflammation [54,55], oxidative stress [56] and many cancers [57]. Additionally, PFKFB3 is a target gene of PPAR-γ and PFKFB3/6-phosphofructo-2-kinase could effectively inhibit intestinal inflammation by activating PPAR-γ [58]. Further investigation revealed that apo-LF could effectively inhibit the LPS-induced reduction of PPAR-γ and PFKFB3 protein expression, effectively alleviating intestinal inflammatory injury (Figure 5). Taken together, the anti-inflammatory effect of apo-LF was inseparable from the inhibition of NF-κB signaling pathway and the activation of PPAR signaling pathway.

In conclusion, this study further explored the potential of apo-LF to strengthen neonatal intestinal immune barrier and proved that apo-LF can effectively inhibit LPS-induced intestinal inflammatory injury in newborns via modulating the NF-κB/PPAR signaling pathway, which provides new insights and reliable scientific basis for the functional study of apo-LF. 

## Figures and Tables

**Figure 1 foods-11-03378-f001:**
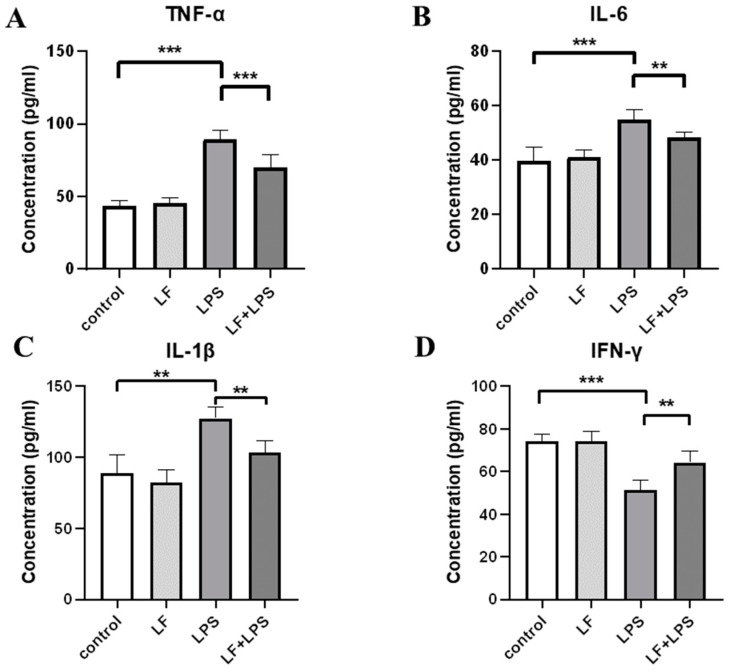
Effect of iron free lactoferrin (apo-LF) on inflammatory cytokines of primary intestinal epithelial cells (IECs) in the co-culture system (primary IECs and immune cells). The concentration of (**A**) pro-inflammatory cytokine TNF-α, (**B**) pro-inflammatory cytokine IL-6, (**C**) pro-inflammatory cytokine IL-1β and (**D**) anti-inflammatory cytokine IFN-γ in the cell supernatant. Lactoferrin (LF) represents apo-LF. Apo-LF represents individual treatment with 10 g/L apo-LF, lipopolysaccharide (LPS) represents individual treatment with 1 mg/L LPS and apo-LF + LPS represents combined treatment with of 10 g/L apo-LF and 1 mg/L LPS. Data analyzed were presented as means ± SD. ** *p <* 0.01, *** *p <* 0.001.

**Figure 2 foods-11-03378-f002:**
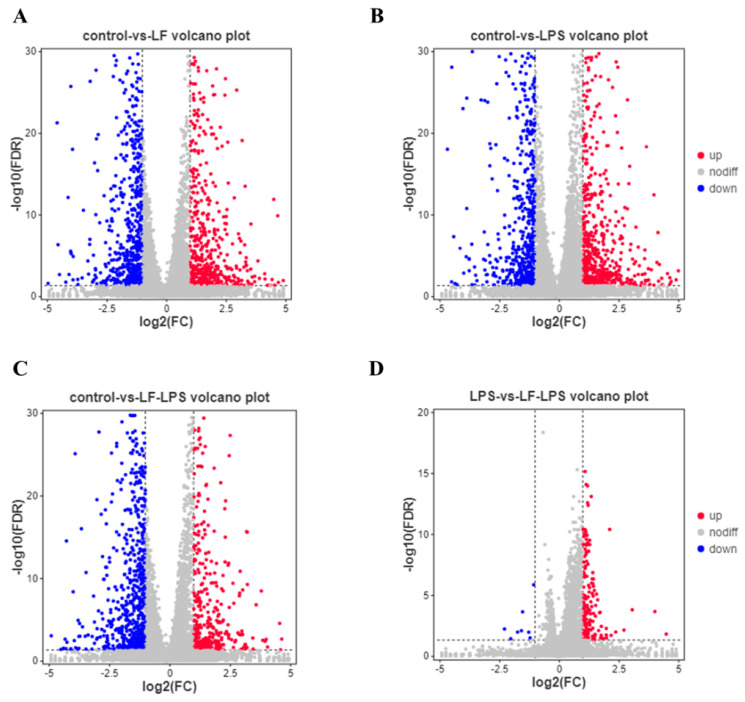
Differential expressed gene (DEGs) analysis of primary IECs in the co-culture system. (**A**–**D**) The number of DEGs screened by group comparison. (**E**) Heatmap of all DEGs. (**F**) Venn analysis between DEGs with different treatments. LF represents apo-LF. Apo-LF represents individual treatment with 10 g/L apo-LF, LPS represents individual treatment with 1 mg/L LPS and apo-LF + LPS represents combined treatment with of 10 g/L apo-LF and 1 mg/L LPS.

**Figure 3 foods-11-03378-f003:**
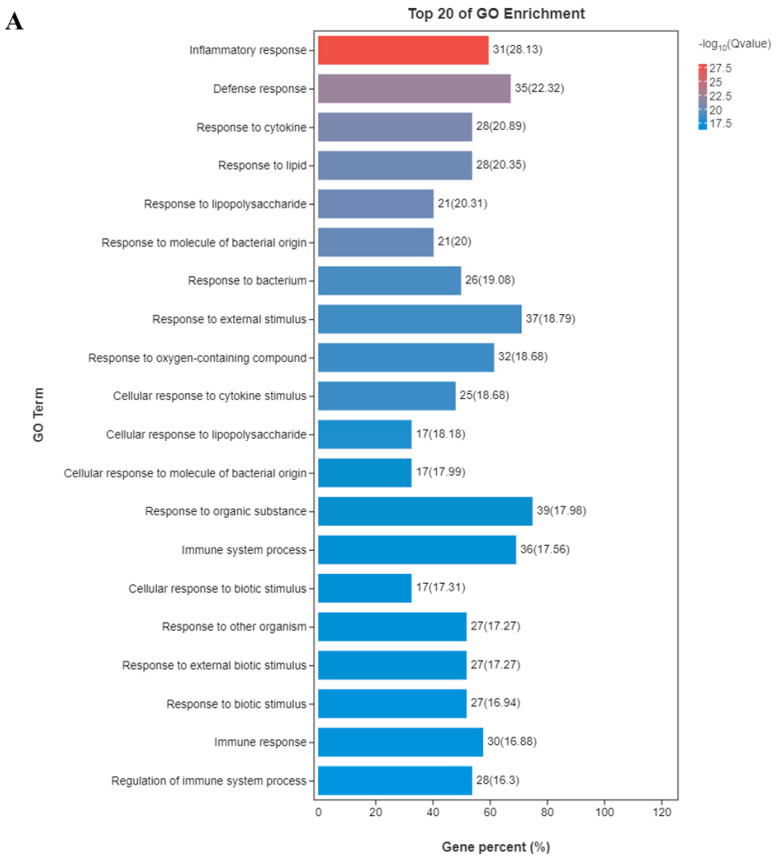
Pathway analysis of primary IECs in the co-culture system. (**A**) Top 20 GO annotations were enriched from 52 DEGs associated with inflammation. (**B**) Top 20 KEGG pathways were enriched from 52 DEGs associated with inflammation. Apo-LF represents individual treatment with 10 g/L apo-LF, LPS represents individual treatment with 1 mg/L LPS and apo-LF + LPS represents combined treatment with of 10 g/L apo-LF and 1 mg/L LPS.

**Figure 4 foods-11-03378-f004:**
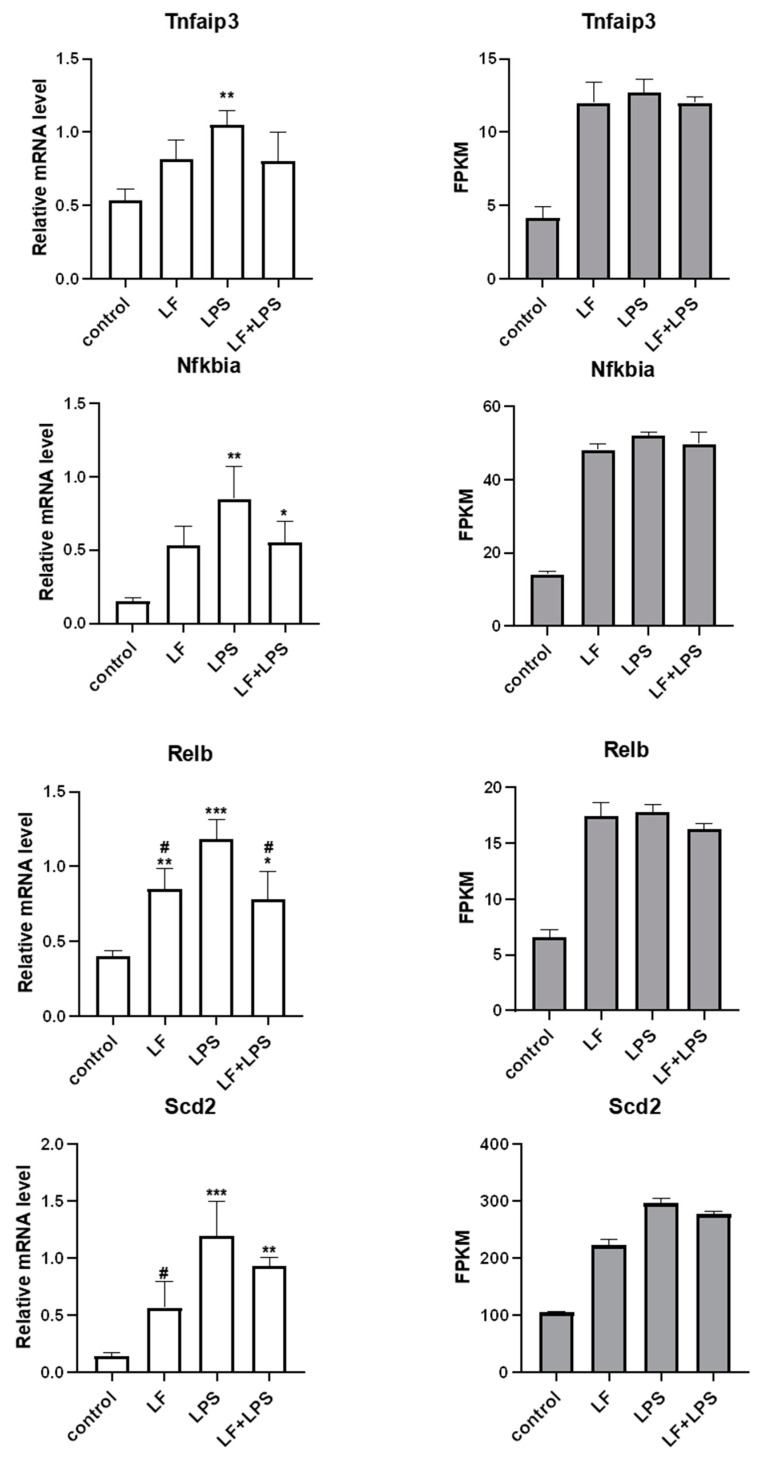
RT-qPCR and RNA-seq were used to determine the expression levels of inflammation-related genes. LF represents apo-LF. Apo-LF represents individual treatment with 10 g/L apo-LF, LPS represents individual treatment with 1 mg/L LPS and apo-LF + LPS represents combined treatment with of 10 g/L apo-LF and 1 mg/L LPS. Data analyzed were presented as means ± SD. * means comparison with the control group, * *p <* 0.05, ** *p <* 0.01, *** *p <* 0.001. **#** means comparison with the LPS group, **#**
*p <* 0.05, **##**
*p <* 0.01, **###**
*p <* 0.001.

**Figure 5 foods-11-03378-f005:**
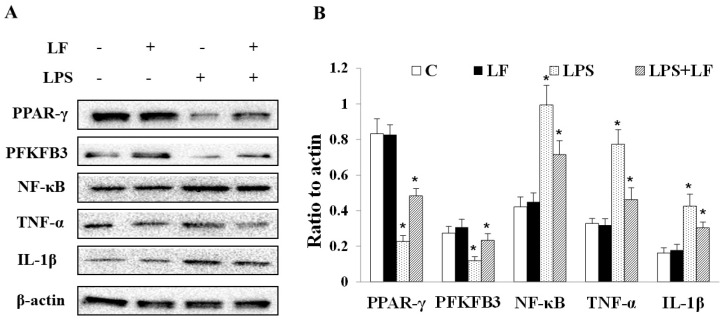
Effects of apo-LF and LPS alone or in combination on inflammation related proteins in colonic tissue of young mice. (**A**) Protein extracts were immunoblotted to capture protein bands. (**B**) Quantification analysis of protein band density. C represents control. LF represents apo-LF. Apo-LF represents individual treatment with 100 mg/kg b.w. apo-LF, LPS represents individual treatment with 10 mg/kg b.w. LPS, LPS + apo-LF represents combined treatment with of 10 mg/kg b.w. LPS and 100 mg/kg b.w. apo-LF. Data analyzed were presented as means ± SD. * means comparison with the control group, * *p <* 0.05.

**Figure 6 foods-11-03378-f006:**
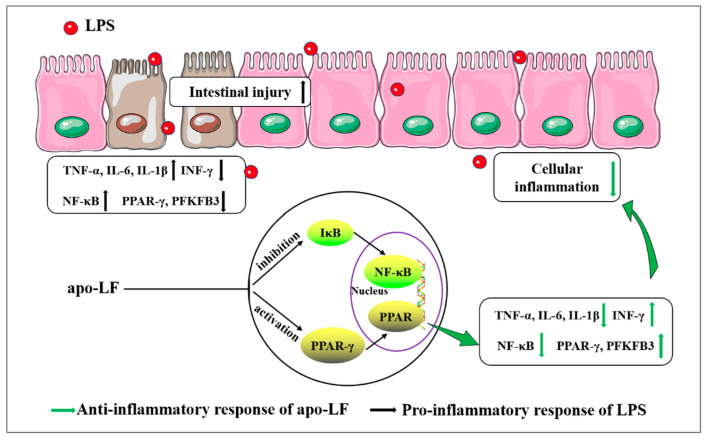
Mechanism diagram of the protective effect of apo-LF on LPS-induced intestinal inflammatory injury.

## Data Availability

Data is contained within the article or Appendix A.

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
