# Peer review of "The Protective Effects of Iron Free Lactoferrin on Lipopolysaccharide-Induced Intestinal Inflammatory Injury via Modulating the NF-κB/PPAR Signaling Pathway"

_foods, 2022, doi:10.3390/foods11213378_

Round 1

Reviewer 1 Report

The manuscript "The protective effects of iron free lactoferrin on lipopolysaccharide-induced intestinal inflammatory injury through the NF-2 κB/PPAR signaling pathway" represents a preclinical study about the potential intestinal anti-inflammatory role that lactoferrin could play through its incorporation into the infant diet. Although further clinical studies will be necessary to support this research, the authors present some first specific experimental steps of a very interesting topic. The manuscript is well written and structured. Sufficient and updated literature is provided. Regarding the experimental part, it is presented sequentially, completely and justified. Furthermore, some minor comments/remarks are given below.

COMMENTS:

The authors show results on primary cultures of IECs from the small intestine, but in the analyzes of the animals they analyze the alterations that occur in the colon of the animals. Why were primary cultures not prepared from colon tissue? Authors should discuss these tissue differences.

In Figure 1 Figure 4, the differences between control and LF+LPS should be analyzed in order to see if there is a complete reversal of the inflammatory process.

The presentation of Figure 4 is not clear. The inclusion of sections (A-F) is not necessary, nor is it commented as such in the manuscript.

The authors should discuss the effect observed in Figure 4, where LF produces a (sometimes significant) increase in all genes related to inflammation that have been analyzed.

The revision of the text by a native English speaker can ameliorate the style and the orthography.

OTHER MINOR COMMENTS:

·         Line 25: add “is” à is a bioactive protein…

·         Line 27: add “In the” à In the study…

·         Line 37: add “suggest that” à “Taken together, suggest that apo-LF

·         Line 45: remove “is” à which is consists of

·         Lines 61-64: please rephrase

·         Line 72: “lactobacillus plantarum” in capital letters and italics à Lactobacillus plantarum

·         Line 136: change “two” by “2” à “thirty-two 2-week.old healthy young male

·         Lines 142-143: specify slaughter method

·         Line 193: delate “studied” or “examined”

·         Lines 322-329: comments made need to be assessed more critically. A short critical comparative discourse between the benefits and harms of using primary cell culture models versus cancer derived cell lines will improve this contribution. This discourse could be based, for example, on whether the lack of morphological differentiation observed in the model used could affect this study, or how a greater differentiation could be promoted.

·         Line 323: delate “primary IECs” à co-cultured primary IECs with immune cells

·         Line 335: insert a space after reference 28

·         Lines 335-342: this phrase might make more sense in other place (such as before line 330)

·         Line 342-343: please, check the sense and the location of the last sentence of this line. Is there preliminary evidence or not?

·         Line 363: ”intestinal?” à maybe you wanted to say “intestine”

·         Line 380-382: please rephrase

Author Response

Dear Reviewer,

Thank you very much for providing an opportunity for us to revise our paper, entitled “The protective effects of iron free lactoferrin on lipopolysaccharide-induced intestinal inflammatory injury via modulating the NF-κB/PPAR signaling pathway” (Manuscript foods-1946901) for the journal. And thank you so much for the constructive comments and valuable suggestions on the manuscript. According to the valuable advices, we have made a revision to this manuscript.

We hope that the manuscript could be acceptable for publication in Foods.

Yours sincerely,

Nan Zheng

Responds to reviewers' comments:

Reviewer #1:

COMMENTS:

Point 1: The authors show results on primary cultures of IECs from the small intestine, but in the analyzes of the animals they analyze the alterations that occur in the colon of the animals. Why were primary cultures not prepared from colon tissue? Authors should discuss these tissue differences.

Response 1: Thanks for your suggestions. We are very sorry for this error, it's our negligence. We have re-verified the materials used and are convinced that the cells were extracted from the colon for the culture. In the following writing, we will pay strict attention to the occurrence of such problems. According to your suggestion, we changed the portion of 2.2 Culture of primary IECs related to small intestine to colon (Line 110-127, page 3).

Point 2: In Figure 1 Figure 4, the differences between control and LF+LPS should be analyzed in order to see if there is a complete reversal of the inflammatory process.

Response 2: Thanks for your suggestions. As your suggestion, we analyzed the difference between the control group and LF+LPS. The details are as follows:

Figure 1 (Line 206-217, page 5): As shown in Figure 1A-C, compared with the control group, the concentrations of pro-inflammatory cytokines, specifically TNF-α, IL-6, and IL-1β in the LPS alone treatment group were significantly increased (P<0.05); after apo-LF intervention, the concentration of the above pro-inflammatory cytokines were still higher than that of the control group, but significantly decreased than that of the LPS group (P<0.05). IFN-γ, an anti-inflammatory cytokine, showed the opposite trend (Figure 1D). Compared with LPS alone group, the concentration of IFN-γ increased significantly after apo-LF and LPS combined treatment (P<0.05), and there was no significant difference compared with the control group. LPS induced cellular inflammation, and apo-LF supplementation in advance could effectively inhibit the inflammatory response induced by LPS, although they would not completely reverse it, which was proved in the above study results.

Figure 4 (Line 288-291, page 9):At the same time, we noted that the mRNA levels of Nfkbia, Relb, Scd2 and Bcl3 were still significantly different between apo-LF + LPS group and control group (P<0.05), indicating that although apo-LF could inhibit LPS-induced inflammatory response, it did not completely reverse.

Point 3: The presentation of Figure 4 is not clear. The inclusion of sections (A-F) is not necessary, nor is it commented as such in the manuscript.

Response 3: Thanks for your suggestions. As your suggestion, we have rearranged Figure 4 for clearer presentation and removed sections (A-F) (Line 292-294, page 10-11).

Point 4: The authors should discuss the effect observed in Figure 4, where LF produces a (sometimes significant) increase in all genes related to inflammation that have been analyzed.

Response 4: Thanks for your suggestions. According to your suggestion, we discussed the three inflammation-related genes (Relb, Bcl3, and Ccl2) in Figure 4 that were significantly different between apo-LF +LPS and LPS group and significantly different between apo-LF and control group. The details are as follows:

(Line 374-395, page 13-14) Relb is one of the more unusual members of the NF-κB family and is known for its role in lymphatic development, dendritic cells biology, and noncanonical signaling [42]. Previous studies have confirmed that Relb overexpression in immune cell lines leads to stronger activation of NF-κB and enhanced LPS-induced production of TNF-α and IL-6 [43].Bcl3 plays a major role in the regulation of immunity and inflammation, promoting or inhibiting the transcription of NF-κB dependent genes, combining with different subunits to exert pro-inflammatory and anti-inflammatory effects [44].LPS stimulation increased Bcl3 expression and accumulation in the nucleus [45].Ccl2 is a chemokine that responds to different inflammatory stimuli, such as LPS and TNF-α, and can prevent a variety of inflammatory diseases by gene deletion and antibody elimination [46].In addition to confirming the reliability of transcriptome data, we found that the mRNA expression levels of the above three inflammation related genes (Relb,Bcl3,and Ccl2) were significantly increased after LPS stimulation (P<0.05), and decreased significantly after apo-LF intervention (P<0.05), which further confirmed the anti-inflammatory effect of apo-LF. Interestingly, we also noticed that compared with the control group, Relb, Bcl3 and Ccl2 were significantly increased in different degrees after apo-LF treatment alone (P<0.05). Previous studies have revealed that cells treated with bovine LF have a strong innate immune response and produce inflammatory reflect compared with the control group [47]. It is speculated that the increased expression of inflammatory genes may be due to the difference of immune regulation of apo-LF, which is affected by various factors such as treatment time and concentration, and mainly plays a related protective role in the stimulation characterized by excessive inflammatory response [48].

Point 5: The revision of the text by a native English speaker can ameliorate the style and the orthography.

Response 5: Thanks for your suggestions. As your suggestion, we have polished our manuscript carefully and corrected the grammatical, styling, and typos found in our manuscript.

OTHER MINOR COMMENTS:

Point 1: Line 25: add “is” à is a bioactive protein…

Response 1: Thanks for your suggestions. As your suggestion, we added “is” in the revised manuscript as ‘Iron free lactoferrin (apo-LF), one of the main types of lactoferrin (LF), is a bioactive protein in milk that plays a vital role in maintaining intestinal health’ (Line 25, page 1).

Point 2: Line 27: add “In the” à In the study…

Response 2: Thanks for your suggestions. As your suggestion, we added “In the” in the revised manuscript as ‘In the study we first explored key genes and pathways in vitro by transcriptome date analysis, and then validated them in vivo to reveal the underlying molecular mechanism’ (Line 27, page 1).

Point 3: Line 37: add “suggest that” à “Taken together, suggest that apo-LF

Response 3: Thanks for your suggestions. As your suggestion, we added “suggest that” in the revised manuscript as ‘Taken together, suggest that apo-LF has a protective effect against LPS-induced intestinal inflammatory injury via modulating the NF-κB/PPAR signaling pathway, which provides new insights for further anti-inflammatory study of apo-LF’ (Line 37, page 1).

Point 4: Line 45: remove “is” à which is consists of

Response 4: Thanks for your suggestions. As your suggestion, we removed “is” in the revised manuscript as ‘Against intestinal pathogens, the mucosal immune system acts as a first line of defense for the body, which consists of closely arranged epithelial cells, immune cells, and their secretions, providing a selective permeable barrier [1]’(Line 45, page 2).

Point 5: Lines 61-64: please rephrase

Response 5: Thanks for your suggestions. As your suggestion, we rephrased in the revised manuscript as ‘Study has demonstrated that LF with different iron saturation has significant differences in inhibiting macrophage inflammation, among which apo-LF has the strongest inhibitory effect [8]’ (Line 64-66, page 2).

Point 6: Line 72: “lactobacillus plantarum” in capital letters and italics à Lactobacillus plantarum

Response 6: Thanks for your suggestions. As your suggestion, we change “lactobacillus plantarum” to capital letters and italic in the revised manuscript as ‘Shin et al. used ileal transcriptome analysis and found that Lactobacillus plantarum strain JDFM LP11 reduced the expression of ileal immune genes on intestinal inflammation and promoted the intestinal development of weaned piglets [14]’ (Line 76, page 2).

Point 7: Line 136: change “two” by “2” à “thirty-two 2-week.old healthy young male

Response 7: Thanks for your suggestions. As your suggestion, we change “two” by “2” in the revised manuscript as ‘Beijing Vital River Laboratory Animal Technology Co., Ltd. (Beijing, China) supplied thirty-two 2-week-old healthy young male C57BL/6J mice (10 ± 3 g)’ (Line 144, page 3).

Point 8: Lines 142-143: specify slaughter method

Response 8: Thanks for your suggestions. As your suggestion, we add slaughter methods in the revised manuscript as ‘The mice were killed by cervical dislocation and their colon tissue was collected for further study on the 15th day’ (Line 151, page 4).

Point 9: Line 193: delate “studied” or “examined”

Response 9: Thanks for your suggestions. As your suggestion, we delate “studied” in the revised manuscript as ‘P<0.05 was examined statistically significant in this study’ (Line 201, page 5).

Point 10: Lines 322-329: comments made need to be assessed more critically. A short critical comparative discourse between the benefits and harms of using primary cell culture models versus cancer derived cell lines will improve this contribution. This discourse could be based, for example, on whether the lack of morphological differentiation observed in the model used could affect this study, or how a greater differentiation could be promoted.

Response 10: Thanks for your suggestions. As your suggestion, we add a brief discussion on the benefits and harms of primary cells culture models and cancer derived cell lines in the revised manuscript as ‘When primary cells leave the body, their biological characteristics are close to those of the original cells, and many of the main features and functions of the original tissue are retained. In contrast, the continuous passage of cancer cells is prone to genetic and phenotypic changes, which are inadequate in accurately reflecting in vivo physiology [30]. Previous studies have shown that primary colonic IECs have a great difference response to inflammation compared with other cancer cell lines studying colitis, as shown in IFN-γ can induce the secretion of IL-8 in Caco-2 cells to remain unchanged, while it increases in HT-29 cells and decreases in primary IECs [31]. This further reminds us of the necessity of using the primary IECs model’ (Line 342-351, page 13).

Point 11: Line 323: delate “primary IECs” à co-cultured primary IECs with immune cells

Response 11: Thanks for your suggestions. As your suggestion, we delate “primary IECs” in the revised manuscript as ‘In the present study, primary IECs was successfully isolated from the colonic tissue of young mice and co-cultured with immune cells (Figure S1-S3)’ (Line 338-339, page 13).

Point 12: Line 335: insert a space after reference 28

Response 12: Thanks for your suggestions. As your suggestion, we insert a space after reference in the revised manuscript as ‘These results were consistent with previous findings in porcine small IECs IPEC-J2 [39] and rat IECs [40]’ (Line 364-365, page 13).

Point 13: Lines 335-342: this phrase might make more sense in other place (such as before line 330)

Response 13: Thanks for your suggestions. As your suggestion, we moved this section to the front in the revised manuscript as ‘Cytokines are widely known to have an irreplaceable effect in the intestinal immune barrier [32]. Some of these cytokines have anti-inflammatory properties (IFN-γ, IL-4 and IL-10), while others have pro-inflammatory properties (TNF-α, IL-6, IL-8 and IL-1β). Regulation of LF on its expression has been supported in a number of in vivo and in vitro studies [33,34]. As is known to all, LPS is an inducer of inflammatory bowel disease and intestinal barrier dysfunction, which can induce morphological, metabolic and gene expression changes of IECs [35,36], leading to uncontrolled expression of host cytokines [37,38]. Subsequently, ELISA method was used to detect cytokines in primary IECs of young mice. It was found that LPS treatment not only significantly induced the secretion of TNF-α, IL-6 and IL-1β, but also lead to a significant decrease in the con-centration of IFN-γ (P<0.05). Apo-LF treatment significantly reversed these trends and alleviated LPS-induced inflammation (Figure 1). These results were consistent with previous findings in porcine small IECs IPEC-J2 [39] and rat IECs [40]’ (Line 353-365, page 13).

Point 14: Line 342-343: please, check the sense and the location of the last sentence of this line. Is there preliminary evidence or not?

Response 14: Thanks for your suggestions. As your suggestion, we have reconsidered the meaning of this sentence and decided to delete it to make the manuscript more rigorous.

Point 15: Line 363: “intestinal?” à maybe you wanted to say “intestine”

Response 15: Thanks for your suggestions. We are sorry for the negligence, and in the revised manuscript, we changed “intestinal” to “intestine” as ‘Previous studies have shown that dietary bovine LF provides limited protection in the intestine by reducing the protein level of pro-inflammatory transcription factor NF-κB and regulating the expression of cytokines TNF-α, IL-1β and IL-10 [52, 53]’ (Line 406-409, page 14).

Point 16: Line 380-382: please rephrase

Response 16: Thanks for your suggestions. As your suggestion, we rephrased in the revised manuscript as ‘In conclusion, this study further explored the potential of apo-LF to strengthen neonatal intestinal immune barrier, and proved that apo-LF can effectively inhibit LPS-induced intestinal inflammatory injury in newborns via modulating the NF-κB/PPAR signaling pathway, which provides new insights and reliable scientific basis for the functional study of apo-LF’ (Line 421-425, page 14).

Reviewer 2 Report

This article deals with an important topic, the use of an important protein like apo-LF to protect against lipopolysaccharide-induced intestinal inflammatory injury. The article used both practical and computational analysis in the investigation. Some issues need corrections.

1.    Title: It is better to clarify the title by changing “through”  to “via modulating” “The protective effects of iron free lactoferrin on lipopolysaccharide-induced intestinal inflammatory injury via modulating the NF- 2 κB/PPAR signaling pathway”.

 2.    Line 60-61 “According to the different amount of iron binding, LF is mainly divided into two types: iron saturated lactoferrin and iron free lactoferrin (apo-LF)”. it is known that there are three forms of lactoferrin, according to its iron saturation: apolactoferrin (iron-free), a monoferric form (containing one ferric ion), and hololactoferrin (containing two iron ions).Please, clarify this part well.

 3.    The introduction section needs more information about the transcriptomics technology.

 4.       Provide the full form of abbreviations at the time of first use, thereafter use the abbreviation, check the article.

 5.       Please review the English and grammar in the article.

 6.    In figure 1, capitalize the first letter in the Y-axis title, and the control “C” in figure 5.

 7.    In figure 4, the graphs with grey bars indicate what?

 8.       Please, rewrite the conclusion.

 9.       You can use these citations to support your discussion:

Author Response

Dear Reviewer,

Thank you very much for providing an opportunity for us to revise our paper, entitled “The protective effects of iron free lactoferrin on lipopolysaccharide-induced intestinal inflammatory injury via modulating the NF-κB/PPAR signaling pathway” (Manuscript foods-1946901) for the journal. And thank you so much for the constructive comments and valuable suggestions on the manuscript. According to the valuable advices, we have made a revision to this manuscript.

We hope that the manuscript could be acceptable for publication in Foods.

Yours sincerely,

Nan Zheng

Responds to reviewers' comments:

Reviewer #2:

Point 1: Title: It is better to clarify the title by changing “through” to “via modulating” “The protective effects of iron free lactoferrin on lipopolysaccharide-induced intestinal inflammatory injury via modulating the NF- 2 κB/PPAR signaling pathway”.

Response 1: Thanks for your suggestions. As your suggestion, we changed “through” to “via modulating” in the revised manuscript as ‘The protective effects of iron free lactoferrin on lipopolysaccharide-induced intestinal inflammatory injury via modulating the NF-κB/PPAR signaling pathway’ (Line 2, page 1).

Point 2: Line 60-61 “According to the different amount of iron binding, LF is mainly divided into two types: iron saturated lactoferrin and iron free lactoferrin (apo-LF)”. it is known that there are three forms of lactoferrin, according to its iron saturation: apolactoferrin (iron-free), a monoferric form (containing one ferric ion), and hololactoferrin (containing two iron ions). Please, clarify this part well.

Response 2: Thanks for your suggestions. As your suggestion, we have redescribed this part in the revised manuscript as ‘According to the different amount of iron binding, LF is mainly divided into three forms: apolactoferrin (apo-LF, iron-free), a monoferric form (containing one iron ion), and hololactoferrin (holo-LF, containing two iron ions). However, due to the limited technology of its separation and extraction, only apo-LF and holo-LF are commercially sold at present’ (Line 60-64, page 2).

Point 3: The introduction section needs more information about the transcriptomics technology.

Response 3: Thanks for your suggestions. As your suggestion, we added more information about the transcriptomics technology in the revised manuscript as ‘Utilizing transcriptomics techniques can achieve an unprecedented level of detail in understanding cellular phenotypes by examining genes expressed in specific physi-ological and pathological states [13]. Currently, the application of transcriptomic technology to explore the effect of functional food supplementation on intestinal in-flammation is very mature. Shin et al. used ileal transcriptome analysis and found that Lactobacillus plantarum strain JDFM LP11 reduced the expression of ileal immune genes on intestinal inflammation and promoted the intestinal development of weaned piglets [14]. Furthermore, colonic transcriptomic analysis of mice revealed that dietary galacto-oligosaccharides could reduce intestinal permeability as well as increase mucus production by modulating changes in host gene expression [15]’ (Line 72-81, page 2).

Point 4: Provide the full form of abbreviations at the time of first use, thereafter use the abbreviation, check the article.

Response 4: Thanks for your suggestions. As your suggestion, we checked the manuscript and revised it according to the correct format (Line 92, page 2; Line 123, page 3).

Point 5: Please review the English and grammar in the article.

Response 5: Thanks for your suggestions. As your suggestion, we have polished our manuscript carefully and corrected the grammatical, styling, and typos found in our manuscript.

Point 6: In figure 1, capitalize the first letter in the Y-axis title, and the control “C” in figure 5.

Response 6: Thanks for your suggestions. As your suggestion, we capitalized the first letter in the Y-axis title, and the control “C” in Figure 5 (Line 218-219, page 5; Line 315, page 12).

Point 7: In figure 4, the graphs with grey bars indicate what?

Response 7: Thanks for your suggestions. The graphs with grey bars indicate “Fragments Per Kilobase of exon model per Million mapped fragments (FPKM)”. It can be obtained by RNA-Seq, which can be used to measure relative gene expression.

Point 8: Please, rewrite the conclusion.

Response 8: Thanks for your suggestions. As your suggestion, we rewrite the conclusion in the revised manuscript as ‘In conclusion, this study further explored the potential of apo-LF to strengthen neonatal intestinal immune barrier, and proved that apo-LF can effectively inhibit LPS-induced intestinal inflammatory injury in newborns via modulating the NF-κB/PPAR signaling pathway, which provides new insights and reliable scientific basis for the functional study of apo-LF’ (Line 421-425, page 14).

Point 9: You can use these citations to support your discussion:

Response 9: As for the citations you mentioned, I don't know why, we haven't seen it in the system. About this part, I will continue to communicate with the editor, so as to timely supplement and modify.

Round 2

Reviewer 1 Report

All comments have been satisfactorily addressed and the manuscript has been correctly improved.